# Prevalence and incidence of sexually transmitted infections among South African women initiating injectable and long-acting contraceptives

Rushil Harryparsad[1], Bahiah Meyer[1], Ongeziwe Taku[1], Myrna Serrano[2], Pai Lien Chen[3], Xiaoming Gao[3], Anna-Lise Williamson[1], Celia Mehou-Loko[1], Florence Lefebvre d'Hellencourt[3], Jennifer Smit[4], Jerome Strauss[5], Kavita Nanda[3], Khatija Ahmed[6,7], Mags Beksinska[4], Gregory Buck[2‡], Charles Morrison[3‡], Jennifer Deese[8‡], Lindi Masson[1,9,10,11‡]*

1 Institute of Infectious Disease and Molecular Medicine (IDM), University of Cape Town, Cape Town, South Africa, 2 Virginia Commonwealth University, Richmond, Virginia, United State of America, 3 FHI 360, Durham, North Carolina, United States of America, 4 MRU, University of the Witwatersrand, Durban, South Africa, 5 Department of Obstetrics and Gynecology, Perelman School of Medicine, University of Pennsylvania, Pennsylvania, PA, United States of America, 6 Setshaba Research Centre, Pretoria, South Africa, 7 Faculty of Health Sciences, Department of Medical Microbiology, University of Pretoria, Pretoria, South Africa8 Pfizer, Inc., Pennsylvania, PA, United States of America, 9 Disease Elimination Program, Life Sciences Discipline, Burnet Institute, Melbourne, Australia, 10 Centre for the AIDS Programme of Research in South Africa, Durban, South Africa, 11 Central Clinical School, Monash University, Melbourne, Australia

‡ GB, CM, JD and LM co-senior authors on this work.
* Lindi.Masson@burnet.edu.au

**Data Availability Statement:** The data that support the findings of this study are available in Table S5.

## Abstract

### Background

South Africa is among the countries with the highest prevalence of sexually transmitted infections (STIs), including *Chlamydia trachomatis* (CT) and *Neisseria gonorrhoeae* (NG). In 2017, there were an estimated 6 million new CT, 4.5 million NG and 71 000 *Treponema pallidum* infections among South African men and women of reproductive age.

### Methods

We evaluated STI prevalence and incidence and associated risk factors in 162 women aged 18–33 years old, residing in eThekwini and Tshwane, South Africa who were part of the Evidence for Contraceptive Options and HIV Outcomes (ECHO) trial. Women were randomised to use depot medroxyprogesterone acetate (n = 53), copper intrauterine device (n = 51), or levonorgestrel (n = 58) implant. Lateral vaginal wall swab samples were collected prior to contraceptive initiation and at months one and three following contraceptive initiation for STI testing.

### Results

There were no significant differences in STI incidence and prevalence across contraceptive groups. At baseline, 40% had active STIs (CT, NG, *Trichomonas vaginalis* (TV),

**Funding:** The research reported in this publication was supported by the Eunice Kennedy Shriver National Institute of Child Health & Human Development of the National Institute of Health under Award Number R01HD096937-01 (PIs: Morrison, Deese and Masson). The content is solely the responsibility of the authors and does not necessarily represent the official views of the National Institutes of Health. LM was supported by the Carnegie Corporation of New York, South African National Research Foundation and Australian National Health and Medical Research Council. The Evidence for Contraceptive Options and HIV Outcomes (ECHO) Trial was supported by the combined generous support of the Bill & Melinda Gates Foundation [grant OPP1032115]; the American people through the United States Agency for International Development [grant AID-OAA-A-15-00045] the Swedish International Development Cooperation Agency [grant 2017/762965-0]; the South Africa Medical Research Council; and the United Nations Population Fund. Contraceptive supplies were donated by the Government of South Africa and US Agency for International Development. The funders had no role in study design, data collection and analysis, decision to publish, or preparation of the manuscript.

**Competing interests:** The authors have declared that no competing interests exist.

*Mycoplasma genitalium* (MG) or herpes simplex virus-2 shedding across all age groups–18–21 years (46%), 22–25 years (42%) and 26–33 years (29%). The incidence of STIs during follow-up was exceptionally high (107.9/100 women-years [wy]), with younger women (18–21 years) more likely to acquire CT (75.9/100 wy) compared to 26–33 year olds (17.4/100 wy; p = 0.049). TV incidence was higher in the 26–33 year old group (82.7/100 wy) compared to the 18–21 year olds (8.4/100 wy; p = 0.01).

## Conclusions

Although the study participants received extensive counselling on the importance of condom use, this study highlights the high prevalence and incidence of STIs in South African women, especially amongst young women, emphasising the need for better STI screening and management strategies.

## Introduction

Sexually transmitted infections (STI) continue to be a global health concern. Most STIs are asymptomatic, often undetected and are therefore left untreated [1–3]. Both asymptomatic and symptomatic STIs can affect fertility, increase risk of human immunodeficiency virus (HIV) infection, and can enhance HIV shedding at genital mucosal sites [4, 5]. South Africa is one of the countries with the highest prevalence of STIs, including *Chlamydia trachomatis* (CT) and *Neisseria gonorrhoeae* (NG), as well as HIV. In 2017, there were an estimated 6 million new CT, 4.5 million NG and 71 000 *Treponema pallidum* infections among South African men and women of reproductive age [6]. STI prevalence generally varies by age, sex and pathogen [6–9]. A 2018 meta-analysis among women in South Africa reported higher CT and NG prevalence among younger (18–24) as compared to older (25–49) women (15% vs. 7.0% and 5% versus 3%, respectively) [9]. On the other hand, syphilis and herpes simplex virus type 2 (HSV-2) were more prevalent in older compared to younger women (4% versus 2% and 78% versus 39%, respectively) and *Trichomonas vaginalis* (TV) prevalence was similar (9% versus 8%) [9]. Among younger women, those aged 20–24 years have been found twice as likely to be infected with a curable STI (CT, NG, syphilis or TV) and HSV-2 when compared to those aged 15–19 years and also more likely to have a STI compared to their male counterparts [4].

Multiple behavioral and social factors increase STI risk among young women including inconsistent or lack of condom use, multiple sexual partners [10, 11], physical and sexual partner violence or coercion, alcohol use and older sexual partners [10–14]. Relationship power inequity, reduced access to sexual health resources [14], and decreased educational and economic opportunities and unemployment [10, 12, 14] also contribute to increased STI risk among young women. Biological factors may also influence a woman's susceptibility to STIs, such as a nonoptimal vaginal microbiome, aberrant inflammation, or decreased protective immunity [15]. Data on whether contraceptive use affects STI acquisition are inconclusive [16, 17], in large part due to the lack of data from prospective and randomized studies. However contraceptive use alters the vaginal microbiome [18–20] and a non-optimal vaginal microbiome has been associated with increased risk of STIs [21]. The Evidence for Contraceptive Options and HIV outcomes (ECHO) trial [22], which evaluated HIV incidence among women randomized to depo-medroxyprogesterone acetate (DMPA-IM), copper-Intrauterine device (copper IUD) or levonorgestrel (LNG) implant, reported 20–28% CT and 4–9% NG

prevalence among South African trial participants at baseline depending on the province [23]. Here, we expand on this previous work by testing for additional STIs, including TV, *Mycoplasma genitalium* (MG) and HSV-2 shedding and evaluate factors associated with STI baseline prevalence and acquisition over three-months following contraceptive randomization.

The aim of the study was to investigate the prevalence and incidence of STIs among South African women and to compare incidence between contraceptive arms, age groups, study sites, and according to condom use and semen detection.

## Methods

### Study participants

The participants in this sub-study were enrolled at the Setshaba Research Centre (SRC) in Tshwane (n = 53) and MatCH Research Unit (MRU) in eThekwini (n = 109) as part of the ECHO trial [22]. ECHO evaluated HIV incidence among 7800 women randomised to DMPA-IM, the LNG implant or the copper-IUD in South Africa, Kenya, Zambia and Eswatini between December 2015 and September 2017 [22]. Women enrolled were nonpregnant, HIV-seronegative, aged 16–35 years, sexually active, seeking effective contraception, had no medical contraindications to the contraceptive methods included in the trial, and had no reported use of injectable, intrauterine or implantable contraception in the previous 6 months. Approximately equal numbers of women in each contraceptive arm were enrolled in this sub-study (51 in the copper IUD group, 53 in the DMPA-IM group, and 58 in the LNG implant group). Demographic, behavioural, and clinical data were collected on standardised case report forms in the parent ECHO trial. Women received HIV and STI risk reduction counselling, HIV testing, syndromic STI treatment and treatment of any diagnosed infection, and were offered male and/or female condoms at each visit. Partner testing and counselling for any study participant was also offered. The University of the Witwatersrand and University of Cape Town Human Research Ethics Committees approved this study and all participants provided written informed consent.

### Specimen collection and processing

For this sub-study, lateral vaginal wall swab samples were collected between June and December 2017 for STI diagnosis and prostate specific antigen (PSA), a biomarker of semen, testing. Specimens were collected at three timepoints—at baseline (immediately before contraceptive method initiation), month one (M1), and month three (M3). Almost all (96%) participants completed all three visits. Samples were collected by placing Dacron swabs on the lateral vaginal wall and rotating 360 degrees, prior to storage at -80˚C for a median time of 23 months (range 21–28 months).

Frozen lateral vaginal wall swabs were thawed on ice overnight at 4˚C prior to elution in 1 mL of phosphate buffered saline (PBS; Sigma-Aldrich, P5493). Tubes were vortexed for 60 seconds and incubated at 4˚C for 1 hour. Excess mucus was then scraped off the inner wall of the tubes and each tube was vortexed again for 30 seconds.

### STI testing

All participants were tested for CT and NG using a nucleic acid amplification test on the GeneXpert instrument system at screening, the final visit and as clinically indicated in the ECHO trial. Additionally, HSV-2 serologic testing was conducted at screening and final visits as part of the parent trial at Bio Analytical Research Corporation South Africa [22]. Syndromic management was provided at screening; participants with positive CT/NG results not treated at

screening were recalled for treatment when results became available. Additional testing for active STIs was conducted as part of the present sub-study as follows. DNA from swab eluants was extracted using the Roche Nucleic acid Kit 1 (Cat. No. 03730964001) and the MagNa Pure Compact Instrument (Product no. 03731146001) and 100μl of the extracted DNA was stored at -20˚C until testing. STI testing was conducted using the STD Direct Flow Chip Kit (Master Diagnostica®—Ref: MAD-003938M-HS12). DNA was amplified using multiplex PCR followed by hybridisation according to the manufacturer's instructions for the detection of multiple STI causing organisms including *Chlamydia trachomatis*, *Neisseria gonorrhoeae*, *Mycoplasma genitalium* and *hominis*, *Ureaplasma urealyticum/parvum*, HSV-1 and 2 and *Treponema pallidum*. The results were analysed using hybriSoft analysis software (Master Diagnostica®). Vaginal swabs were also tested for the presence of *T. vaginalis* as described by Schirm *et al.* [24]. Primers and JOE-labeled probe targeting *T. vaginalis*-specific 2-kb repeated sequence was employed using ViiA 7 Real-Time PCR System (ThermoFisher Scientific). Samples were processed in duplicate and those with mean cycle threshold (Ct) values <40 were considered positive. Wells with no DNA served as negative controls, and standard curves using serial dilutions of *T. vaginalis* genomic DNA were included.

## PSA measurement

Swab eluants were transferred into filter centrifuge tubes (Corning® Costar® Spin-X® tubes Sigma-Aldrich, CLS8160) and centrifuged for 10 min at 4000 Relative Centrifugal Force (RCF). Filters were removed and the supernatants vortexed for 10 seconds at a low speed. PSA as a marker of recent unprotected sex was measured using Human Kallikrein 3/PSA Quantikine ELISA (R&D Systems, US–catalog no. DKK300). Reagent preparation and assay procedures were conducted according to the manufacturer's instructions. Samples were analysed on an ELISA plate reader Spectramax 250 for PSA at 450nm.

## Data analysis

Statistical analyses were performed using GraphPad Prism (GraphPad Software, USA) and R Studio (R Studio Software, USA). Women were grouped by age to include approximately equal numbers within each group (18–21; 22–25; 26–33 years). We used the Fisher exact test to compare baseline STI prevalence between groups, including age group, contraceptive group (copper-IUD, LNG implant and DMPA-IM) and Chi-Square test for PSA detection (detected/not detected), condom use (yes/no) and site (eThekwini/Tshwane). Kruskal-Wallis test was used to compare weight, height, body mass index and number of sex acts in the past three months. We compared STI incidence over the three-month follow-up period between groups using the Kaplan-Meier method. P-values were adjusted for multiple testing using a Bonferroni correction [25]. Incidence was defined as the total number of new cases/ total person-time at risk and prevalence was defined as the total number of affected individuals/ total number of individuals in the population.

## Results

### Cohort characteristics

Baseline demographic, behavioural and clinical characteristics of the study participants are described by age group (Table 1) and study site (S1 Table). As previously reported, baseline characteristics of the women included did not differ by contraceptive group [26]. Older women (26–33 years) reported significantly more frequent sexual activity and had higher BMI compared to the younger women (18–21 and 22–25 years; p<0.001) and Tshwane participants

**Table 1. Baseline demographic, behavioural, and clinical characteristics of study participants overall and by age group.**

| | Total (n = 162) | Age: 18–21 (n = 52) | Age: 22–25 (n = 62) | Age: 26–33 (n = 48) | p-value* |
|---|---|---|---|---|---|
| **Education [n(%)]** | | | | | 0.39 |
| Secondary | 126 (78) | 41 (79) | 45 (73) | 40 (83) | |
| Post-secondary | 36 (22) | 11 (21) | 17 (27) | 8 (17) | |
| **Weight (kg)** | 68.9 | 62.3 | 70.0 | 74.8 | **<0.001*** |
| **[mean(range)]** | (40.7–135.2) | (41.1–104.6) | (40.7–113.3) | (47.2–135.2) | |
| **Height (cm)** | 158.6 | 158.6 | 158.40 | 158.9 | 0.84 |
| **[mean(range)]** | (143–188) | (144–188) | (143–173) | (149–172) | |
| **Body mass index** | 27.4 | 24.7 | 27.9 | 29.6 | **<0.001*** |
| **[mean(range)]** | (17.2–52.2) | (17.9–40.9) | (17.2–46.0) | (47.2–52.2) | |
| **Vaginal intercourse in past 3** | 16 | 13 | 14 | 22 | **<0.001*** |
| **months [mean(range)]** | (0–60) | (1–48) | (0–50) | (1–60) | |
| **Marital status [n(%)]** | | | | | 0.44 |
| Married | 1 (1) | 0 (0) | 1 (2) | 0 (0) | |
| Not Married | 161 (99.4) | 52 (100) | 61 (98.4) | 48 (100) | |
| **Current smoker [n(%)]** | 37 (23) | 7 (14) | 16 (26) | 14 (29) | 0.14 |
| **Prostate specific antigen detected [n(%)]** | 21 (13) | 6 (12) | 9 (15) | 6 (13) | 0.89 |
| **Ever use condoms [n(%)]** | | | | | 0.61 |
| Yes | 69 (43) | 20 (39)[¥] | 26 (42) | 23 (49)[¥¥] | |
| No | 91 (56) | 31 (61)[¥] | 36 (58) | 24 (51)[¥¥] | |

Abbreviations

[¥] n = 51

[¥¥] n = 47; kg, kilogram; cm, centimeter. Proportions were compared using the Chi-Square test and continuous variables were compared using Kruskal-Wallis test.

*p<0.05 following Bonferroni correction was considered statistically significant.

reported significantly higher sexual frequency than eThekwini participants (S1 Table). Women at the Tshwane site were less likely to have completed post-secondary education compared to women enrolled at eThekwini.

## Baseline STI prevalence

STI prevalence was high overall, with 40% of women having an active STI (CT, NG, TV, MG or HSV-2 shedding), 10% having more than one STI, 19% having CT and 19% having TV (Table 2). The prevalence of any active STI was highest in younger women [18–21 years (46%), 22–25 years (42%), 26–33 years (29%); Fig 1]. A significantly higher prevalence of CT was noted in 18–21 year old women (29%) compared with 26–33 year old women (8%; p = 0.01). HSV-2 seropositivity was higher in 26–33 and 22–25 year age groups (53% and 44% respectively), compared to the 18–21 year age group (16%; p<0.001 and p<0.01 respectively). The overall prevalence of active STIs did not differ significantly between sites [eThekwini (36%) and Tshwane (47%); p = 0.17; S2 Table; S1 Fig], however women enrolled at the Tshwane site were almost three times more likely to have TV compared to eThekwini (p<0.01). No significant differences in STI prevalence between randomized contraceptive groups before contraceptive initiation were noted (S3 Table).

## STI incidence

The incidence of any active STI during the three-month follow-up period was very high [107.9/100 women years (wy); Table 3], with a trend toward increasing incidence among older

**Table 2. Baseline STI prevalence overall and by age group.**

| STI | Total (n = 162) n (%) | Age Group n (%) | | | p-value* | | |
| --- | --- | --- | --- | --- | --- | --- | --- |
| | | 18–21 years (n = 52) | 22–25 years (n = 62) | 26–33 years (n = 48) | 18–21 vs 22–25 years | 18–21 vs 26–33 years | 22–25 vs 26–33 years |
| Any active STI^* | 64 (40) | 24 (46) | 26 (42) | 14 (29) | 0.71 | 0.10 | 0.23 |
| *Chlamydia trachomatis* | 31 (19) | 15 (29) | 12 (19) | 4 (8) | 0.27 | **0.01*** | 0.17 |
| *Neisseria gonorrhoeae* | 7 (4) | 3 (6) | 3 (5) | 1 (2) | >0.99 | 0.62 | 0.63 |
| *Trichomonas vaginalis* | 31 (19) | 10 (19) | 13 (21) | 8 (17) | >0.99 | 0.80 | 0.63 |
| *Mycoplasma genitalium* | 7 (4) | 1 (2) | 4 (6) | 2 (4) | 0.37 | 0.61 | 0.69 |
| **Herpes simplex virus 2 shedding** | 9 (6) | 3 (6) | 5 (4) | 1 (2) | 0.73 | 0.62 | 0.42 |
| **Herpes simplex virus 2 serology** | 54 (38) | 7 (16) | 24 (44) | 23 (53) | **<0.01*** | **<0.001*** | 0.23 |
| **Multiple active STIs** | 16 (10) | 5 (7) | 9 (15) | 2 (4) | 0.57 | 0.44 | 0.11 |

Abbreviations: STI, sexually transmitted infection.

^tested positive for at least one STI (CT, NG, TV, MG or HSV-2). Proportions were compared using the Fishers Exact test

*p<0.05 following Bonferroni correction was considered statistically significant.

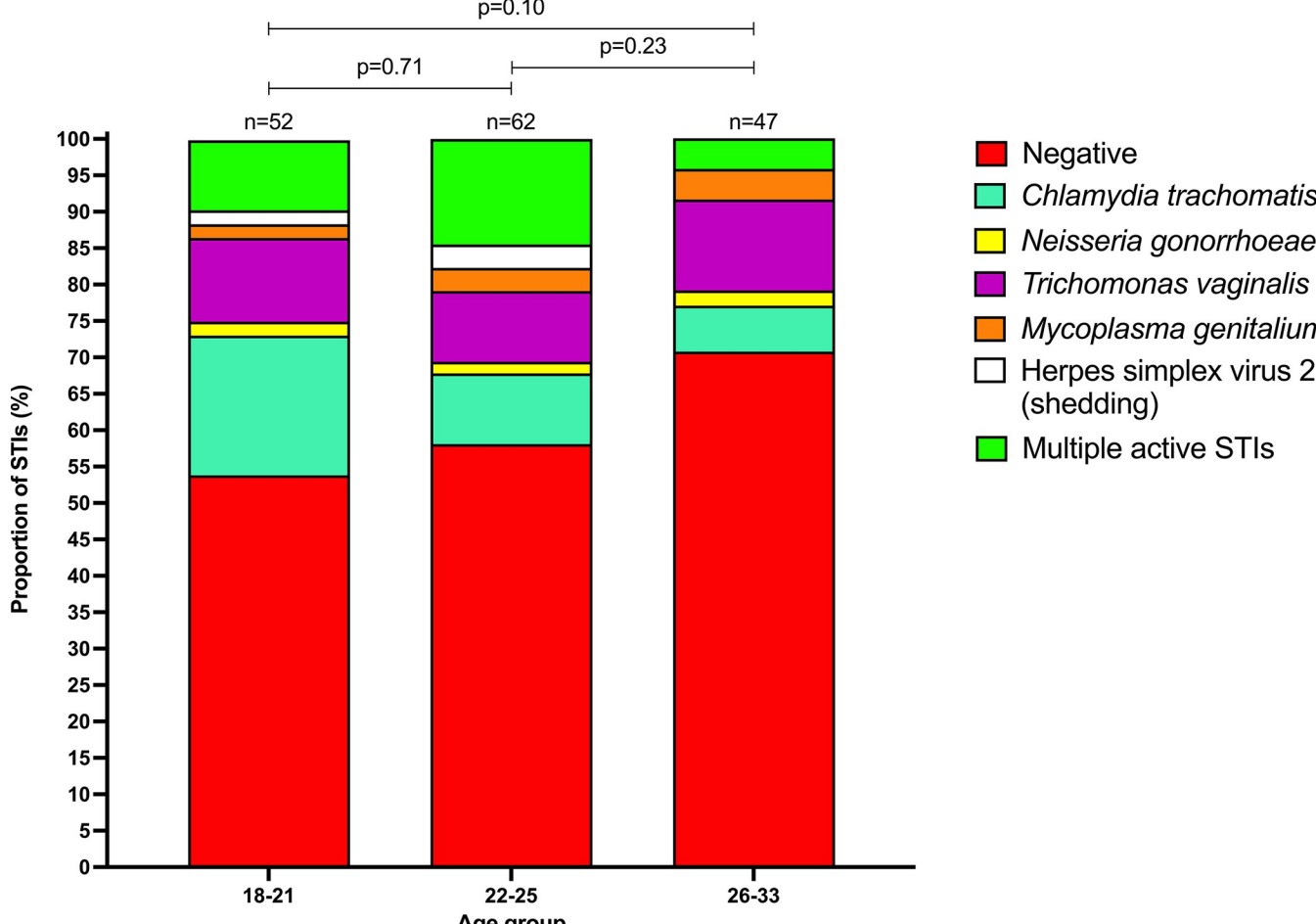

**Fig 1. Total proportion of women with active sexually transmitted infections (STIs) at baseline by age group.** The younger (18–21, n = 52) and middle (22–25, n = 62) age group had a greater proportion of STI infections than the older age group (26–33, n = 47). The younger age group had a greater proportion of *Chlamydia trachomatis* (CT) infections compared to the other two groups, the middle age group had a higher proportion of multiple STIs compared to the younger and older age groups and the older age group had a higher proportion of *Trichomonas vaginalis* (TV) infections.

**Table 3. Three-month STI incidence overall and by contraceptive group.**

| STI | Total (n/100wy) | Contraceptive Arms (n/100wy) | | | | p-value* | | |
| --- | --- | --- | --- | --- | --- | --- | --- | --- |
| | | LNG implant | DMPA-IM | Cu-IUD | p-value | LNG implant vs DMPA-IM | LNG implant vs Cu-IUD | DMPA-IM vs Cu-IUD |
| **Any active STI** | 107.9 | 72.5 | 143.82 | 114.2 | 0.28 | 0.11 | 0.32 | 0.57 |
| *Chlamydia trachomatis* | 40.4 | 16.8 | 64.5 | 42.2 | 0.20 | 0.07 | 0.24 | 0.50 |
| *Neisseria gonorrhoeae* | 7.7 | 0 | 7.8 | 16.5 | 0.31 | 0.29 | 0.13 | 0.54 |
| *Trichomonas vaginalis* | 43.7 | 42.2 | 54.7 | 37.5 | 0.82 | 0.66 | 0.84 | 0.55 |
| *Mycoplasma genitalium* | 18.0 | 22.3 | 23.8 | 7.8 | 0.58 | 0.92 | 0.36 | 0.31 |
| **Multiple active STIs** | 35.5 | 14.9 | 32.4 | 64.7 | 0.13 | 0.33 | 0.05 | 0.29 |

*$p < 0.05$ following Bonferroni correction was considered statistically significant.

women (100.6/100wy, 103.8/100 wy and 121.3/100 wy for 18–21, 22–25 and 26–33 year age groups, respectively). However, CT incidence was higher among women aged 18–21 years (75.9/100 wy) compared to those aged 22–25 and 26–33 years (31.6/100 wy and 17.4/100 wy respectively), with a significant difference between the 18–21 and 26–33 year age groups (p = 0.049; Fig 2B). In contrast, STI incidence was higher for TV in the 26–33 year age group (82.7/100 wy) compared to the 18–21 and 22–25 year age groups (8.4/100 wy and 50.5/100 wy respectively), with a significant difference between the 18–21 and 26–33 year age groups (p = 0.01; Fig 2D). Although STI incidence between sites was relatively similar, there was a nonsignificant trend towards a greater incidence of multiple active STIs at the eThekwini site compared to the Tshwane site (48.9/100wy and 8.3/100wy respectively; S4 Table, p = 0.06). When the data were stratified by age group, 26–33 year old women at eThekwini were more likely to acquire a STI compared to women of the same age at Tshwane (p = 0.002; S2 Fig).

No statistically significant differences in STI incidence between contraceptive groups was noted, but trends towards higher incidence of any active STI or CT alone were observed in the DMPA-IM arm compared to LNG implant users (p = 0.11 and p = 0.07, respectively; Table 3). Incidence of multiple concurrent STIs was higher for copper IUD and DMPA-IM than LNG implant users (64.7/100 wy, 32.4/100 wy, and 14.9/100 wy respectively, p = 0.13). No statistically significant relationships were noted between the incidence of an active STI incidence and PSA detection (S3 Fig). PSA detection was not associated with reported condom use (S4 Fig), but was significantly associated with reported vaginal intercourse between 0–4 (38%) and 8–13 (12%) days before the study visit (p = 0.03; S5 Fig).

## Discussion

STIs have a major impact on both sexual and reproductive health and, in addition to their individual pathogenic effects, increase the risk of HIV, adverse pregnancy outcomes and infertility. Understanding factors that influence STI prevalence and incidence is critical for the implementation of effective management strategies to reduce the burden of these infections and associated adverse outcomes. Similar to previous studies [3, 4, 6, 9], our study found a high burden of STIs in South African women, with 40% of women having an active STI, 19% having CT, 29% TV, 38% HSV-2 seropositivity and 10% having more than one active STI. During the three-month follow-up period, we observed a high incidence (107.9/100wy) of any active STI.

Younger women (18–21 years) had a higher prevalence of CT compared to older women (26–33 years), 29% vs 8% respectively. This is similar to the prevalence reported in the ECHO parent study that recorded CT prevalence of 20–28% in South African women at baseline and

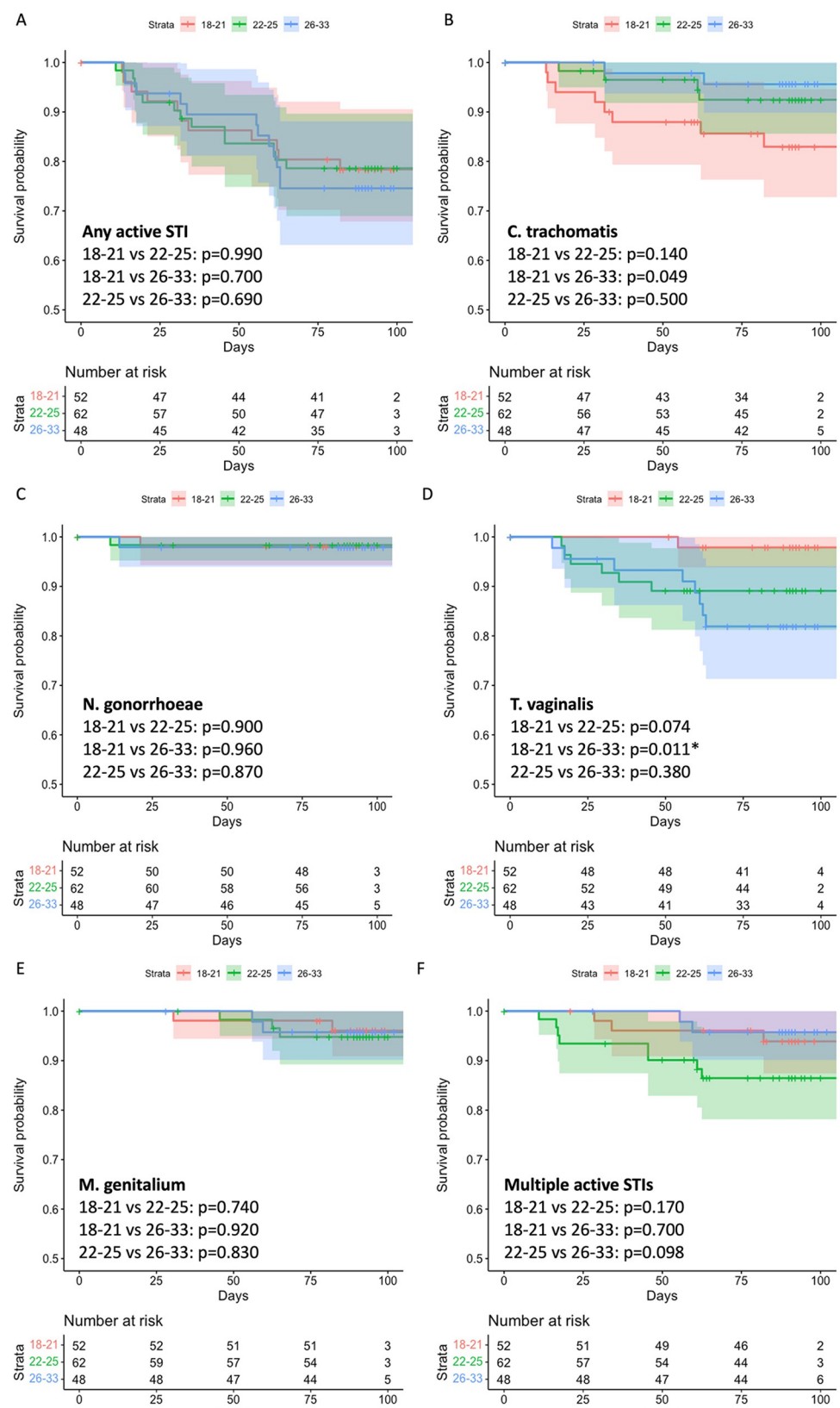

**Fig 2.** (A-F), Kaplan-Meier curve showing sexually transmitted infection (STI) incidence between age groups over time (days). Red represents the 18–21 year olds, green the 22–25 year olds and blue the 26–33 year olds. The tables show the number of individuals at risk of acquiring STIs over time. A vertical drop represents an event has occurred and a verticle tick mark on the curves indicate that a participant has been censored at that timepoint. (A) Any STI incidence over time. There were no significant differences between any age groups. (B) *Chlamydia trachomatis* (CT) incidence over time. There was a significant difference between 18–21 and 26–33 year olds (p = 0.049). (C) *Neisseria gonorrhoeae* (NG) incidence over time. There were no significant differences between any age groups. (D) *Trichomonas vagnalis* (TV) incidence over time. There was a signigficant difference between the 18–21 and 26–33 year olds (p = 0.011*). (E) *Mycoplasma genitalium* (MG) incidence over time. There were no significant differences between any age groups. (F) Multiple STI incidence over time. There were no significant differences between age groups. *p<0.05 following Bonferroni correction was considered statistically significant.

found that younger women were more likely to have a CT infection than older women [23]. In contrast, and as expected, HSV-2 seropositivity was higher in the older age groups (22–25 and 26–33; 44% and 53% respectively) compared to the younger age group (18–21; 16%). Similar to the prevalence data for 18–21 year-olds, this age group was more likely to acquire CT during the follow-up period while older women aged 26–33 were more likely to acquire TV. Previous studies have hypothesized that long-lasting partial immunity to CT may develop [27] and this may explain the difference in prevalence and incidence observed between age groups. On the other hand, more frequent sexual activity reported by older women could account for the greater incidence of TV. We did not find any statistically significant differences in STI incidence between contraceptive arms, but there was a trend towards higher incidence of CT and any STI in the DMPA-IM arm compared to the LNG implant users through three months. In contrast, an analysis of CT and NG prevalence following 18 months of contraceptive use in the ECHO parent study, which included all clinical sites, found that CT was less prevalent in women using DMPA-IM compared to LNG implant users [23]. This difference between our analyses and that of the parent study may be explained by socio-behavioral differences between the women enrolled at different clinical sites in South Africa and other countries. For example, reported sexual activity differed between Tshwane and eThekwini and TV prevalence at the Tshwane site was almost three times that of eThekwini while other STI infections were similar between sites. CT and NG prevalence rates at the eThekwini study site in Kwa-Zulu Natal (21% and 3% respectively) were comparable to the prevalence reported in this province in the ECHO parent study [23], however, CT prevalence was double in the Gauteng province overall compared to the Tshwane study site in this study (20% vs 10% respectively). Additionally, other behavioral factors may play a role, including differences in smoking, educational level or clinical factors such as cervical ectopy. Additionally, contraceptive-associated risk may change with use over time with long-acting methods, such as decreasing LNG implant release rates over time.

We measured PSA as a biomarker of semen to evaluate the relationships between recent unprotected sexual intercourse, reported condom use and STI incidence. Only 13% of individuals tested positive for PSA at baseline and, while PSA was not associated with reported condom use, it was significantly associated with reporting of recent vaginal sexual acts. We did not find any association between PSA positivity across all visits and STI incidence, and we did not observe significant differences in PSA detection between age groups, study sites or contraceptive arms. However, a study by Deese et al. [28], investigating PSA detection frequency in another sub-study of the ECHO clinical trial showed that PSA was detected less frequently in women randomised to DMPA-IM compared to copper-IUD and LNG implant. These data suggest that women using copper-IUD and LNG implant may have had condomless vaginal sex more frequently than women using DMPA-IM. Other studies have shown that PSA is associated with bacterial vaginosis prevalence and is involved in cytokine changes in the female genital tract [29–31].

This study had several important strengths including randomization to contraceptive method, high adherence to contraceptive method and testing for STIs and other co-factors. However, limitations include the small sample size, short follow-up period, and limited power to evaluate differences between some of the STIs between the different groups. It would also be vital to study the STI incidence and prevalence rates in women living with HIV as this cohort included only seronegative women. This has importance as previous studies have reported a bidirectional relationship between HIV and STIs, with women with HIV more likely to have a STI compared to HIV-negative women [32] and STIs associated with increased risk of HIV acquisition [33]. Overall, this study adds to the limited data available on contraceptive use and STI acquisition [34].

In conclusion, this study found a high prevalence and incidence of STIs, especially in young women who are also most at risk of HIV infection in South Africa [4, 6, 9, 23, 35]. These findings highlight the need for better STI diagnosis and management in this setting where STIs are only treated if women present with signs or symptoms [2], even though most infections are asymptomatic and remain untreated [3]. Other factors that may contribute to high STI prevalence include limited education about STIs and their prevention, highlighting the need to empower the youth by educating them on sexual and reproductive health as well as encouraging proper use of condoms. The introduction of low-cost point-of-care (POC) STI testing in these settings will be of critical importance to effectively manage these infections. A recent study in South Africa found that POC testing, expedited partner therapy and immediate treatment of women with STIs successfully reduced STI prevalence after 12 weeks [36]. A South African research team is currently working on a Genital Inflammation Test (GIFT) for identifying women with asymptomatic STIs and bacterial vaginosis who are at risk for HIV acquisition [37, 38]. This intervention in conjunction with syndromic management may help lower the STI burden among South African women.

## Supporting information

**S1 Questionnaire. Inclusivity in global research.**
(DOCX)

**S1 Fig. Total proportion of women with active sexually transmitted infections (STIs) at baseline by study site.** The Setshaba (n = 53) study site had a greater proportion of *Trichomonas vaginalis* (TV) and multiple STI infections compared to the MatCH (n = 109) study site.
(JPG)

**S2 Fig.** (A-C). Kaplan-Meier curve showing sexually transmitted infection (STI) incidence between study site by age group over time (days). Red represents the MatCH site, green the Setshaba site. The tables show the number of individuals at risk of acquiring STIs over time. A vertical drop represents an event has occurred and a verticle tick mark on the curves indicate that a participant has been censored at that timepoint. (A), STI incidence between sites for 18–21 year olds were not statistically significant. (B), There was no statistically significant differences for 25–33 years olds between sites. (C), STI incidence between sites for 26–33 year olds had a statistically significant difference (p = 0.002).
(JPG)

**S3 Fig. Kaplan-Meier curve showing any active sexually transmitted infection (STI) incidence between participants testing positive or negative for vaginal prostate specific antigen (PSA).** Red represents participants positive for PSA at any visit, green indicates participants negative for PSA at all visits. The tables show the number of individuals at risk of acquiring any active STI over time. A vertical drop represents an event has occurred and a cross mark on

the curves indicates that a participant has been censored at that timepoint. There was no statistically significant difference in the incidence of an active STI between participants positive or negative for PSA.
(JPG)

**S4 Fig. Total proportion of women reporting ever using condoms according to prostate specific antigen (PSA) detection at baseline.** There were no significant differences in PSA detection in women who reported condom use and in women reporting no condom usage.
(JPG)

**S5 Fig. Vaginal prostate specific antigen (PSA) concentrations according to reported time since last vaginal intercourse (days).** PSA concentration was significantly higher among women reporting vaginal intercourse 0–4 days compared to 8–13 days prior to the study visit (p = 0.03).
(JPG)

**S1 Table. Baseline demographic, behavioural and clinical characteristics of study by study site.** Abbreviations: ¥ n = 107;kg, kilogram; cm, centimeter. Proportions were compared using the Chi-Square test and continuous variables were compared using Kruskal-Wallis test. *p<0.05 following Bonferroni correction was considered statistically significant.
(DOCX)

**S2 Table. Baseline STI prevalence overall and by study site.** Abbreviations: STI, sexually transmitted infection. ^tested positive for at least one STI (CT, NG, TV, MG or HSV-2). Proportions were compared using the Fishers Exact test *p<0.05 following Bonferroni correction was considered statistically significant.
(DOCX)

**S3 Table. Baseline STI prevalence overall and by contraceptive group.** Abbreviations: STI, sexually transmitted infection. ^tested positive for at least one STI (CT, NG, TV, MG or HSV-2). Proportions were compared using the Fishers Exact test *p<0.05 following Bonferroni correction was considered statistically significant.
(DOCX)

**S4 Table. Three-month STI incidence overall and by study site.** *p<0.05 following Bonferroni correction was considered statistically significant.
(DOCX)

**S5 Table. Full study dataset.**
(XLSX)

## Acknowledgments

We thank all the women who participated in this study, the communities that supported this work and the study teams responsible for the collection, processing, storage and shipping of the samples. The contents of this paper are solely the responsibility of the authors and do not necessarily reflect the views, decisions, or policies of the institutions with which they are affiliated, the ECHO trial funders, or the supporting governments.

## Author Contributions

**Conceptualization:** Charles Morrison, Jennifer Deese, Lindi Masson.

**Data curation:** Rushil Harryparsad, Xiaoming Gao.

**Formal analysis:** Rushil Harryparsad, Bahiah Meyer, Myrna Serrano, Pai Lien Chen, Xiaoming Gao.

**Funding acquisition:** Kavita Nanda, Charles Morrison, Jennifer Deese, Lindi Masson.

**Investigation:** Rushil Harryparsad, Bahiah Meyer, Ongeziwe Taku, Myrna Serrano, Jennifer Smit, Khatija Ahmed, Mags Beksinska.

**Methodology:** Ongeziwe Taku, Myrna Serrano, Jennifer Smit.

**Project administration:** Celia Mehou-Loko, Florence Lefebvre d'Hellencourt, Mags Beksinska, Charles Morrison, Jennifer Deese, Lindi Masson.

**Resources:** Jennifer Deese, Lindi Masson.

**Supervision:** Ongeziwe Taku, Pai Lien Chen, Anna-Lise Williamson, Celia Mehou-Loko, Florence Lefebvre d'Hellencourt, Jennifer Smit, Jerome Strauss, Kavita Nanda, Khatija Ahmed, Gregory Buck, Charles Morrison, Jennifer Deese, Lindi Masson.

**Validation:** Rushil Harryparsad, Myrna Serrano, Anna-Lise Williamson, Jerome Strauss, Charles Morrison, Lindi Masson.

**Visualization:** Rushil Harryparsad, Gregory Buck.

**Writing – original draft:** Rushil Harryparsad.

**Writing – review & editing:** Bahiah Meyer, Ongeziwe Taku, Myrna Serrano, Pai Lien Chen, Xiaoming Gao, Anna-Lise Williamson, Celia Mehou-Loko, Florence Lefebvre d'Hellencourt, Jennifer Smit, Jerome Strauss, Kavita Nanda, Khatija Ahmed, Mags Beksinska, Gregory Buck, Charles Morrison, Jennifer Deese, Lindi Masson.

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
