## [Decision Letter · Decision Letter 0]

9 Jul 2023

PONE-D-23-13490Prevalence and incidence of sexually transmitted infections among South African women initiating injectable and long-acting contraceptivesPLOS ONE

Dear Dr. Masson,

Thank you for submitting your manuscript to PLOS ONE. After careful consideration, we feel that it has merit but does not fully meet PLOS ONE’s publication criteria as it currently stands. Therefore, we invite you to submit a revised version of the manuscript that addresses the points raised during the review process.

We look forward to receiving your revised manuscript.

Kind regards,

Rebecca F. Baggaley

Academic Editor

PLOS ONE

Journal Requirements:

“The research reported in this publication was supported by the Eunice Kennedy Shriver National Institute of Child Health & Human Development of the National Institute of Health under Award Number R01HD096937-01 (PIs: Morrison, Deese and Masson). The content is solely the responsibility of the authors and does not necessarily represent the official views of the National Institutes of Health. LM was supported by the Carnegie Corporation of New York, South African National Research Foundation and Australian National Health and Medical Research Council. The Evidence for Contraceptive Options and HIV Outcomes (ECHO) Trial was supported by the combined generous support of the Bill & Melinda Gates Foundation [grant OPP1032115]; the American people through the United States Agency for International Development [grant AID-OAA-A-15-00045] the Swedish International Development Cooperation Agency [grant 2017/762965-0]; the South Africa Medical Research Council; and the United Nations Population Fund. Contraceptive supplies were donated by the Government of South Africa and US Agency for International Development.”

6. Please amend your manuscript to include your abstract after the title page.

Reviewers' comments:

Reviewer's Responses to Questions

**Comments to the Author**

1. Is the manuscript technically sound, and do the data support the conclusions?

Reviewer #1: Partly

Reviewer #2: Yes

Reviewer #3: Yes

2. Has the statistical analysis been performed appropriately and rigorously? 

Reviewer #1: No

Reviewer #2: Yes

Reviewer #3: Yes

3. Have the authors made all data underlying the findings in their manuscript fully available?

Reviewer #1: No

Reviewer #2: Yes

Reviewer #3: No

4. Is the manuscript presented in an intelligible fashion and written in standard English?

Reviewer #1: Yes

Reviewer #2: Yes

Reviewer #3: Yes

5. Review Comments to the Author

Reviewer #1: This is an interesting study finding an alarmingly high prevalence and incidence of STIs among a population highly vulnerable to HIV. I think changes need to be made to the structure and the methodology before it is published. I will gladly review it again more thoroughly once my main concerns have been addressed. These are:

1. I find it unclear what the aim of the analysis is. Is it to estimate STI prevalence and incidence in this population, or is it to explore the possible relationship between type of contraceptive and STI incidence? Please make the aim clear, explicitly with an aim statement at the end of the Introduction and implicitly throughout the narrative of the paper.

2. I would request that a statistician also review the paper. I have a number of concerns:

a. Incidence should be defined

b. The reason for using keplan meier and the benefit of that approach is unclear to me

c. The reason for plotting the age groups separately is also unclear. Is it to test a hypothesis that incidence would differ by age? The age categories seem somewhat arbitrary. How were they decided upon?

General language suggestions:

There are too many accronyms. As a minimum, I suggest referring to the MRU site as the eThekwini site instead, and the SRC as the Tschwane site.

Define wy when first used

I’m not sure this makes sense: ‘seeking effective contraception with no medical contraindications to trial contraceptive methods’

Reviewer #2: Very good piece of research. I have a few minor comments.

In the introduction I would reccomend changing the statements to "can cause shedding" and "can affect fertility" as it is not true in 100 percent of cases. I also reccomend splitting up the long list of reasons why women may be more susceptible to acquiring an sti. The list is long and the references need to match the risk factors presented. Also categories such as "gender inequality" is too broad. It is important to be more specific as to how these risks arise and why they are specific to young women.

There are some formatting issues in the table where the rows are not vertically aligned. And I think each table needs a description of the p-value definition. Particularly table 3 which has a "P-value" column alongside three columns under the heading "p-value". Descriptions in the text or table caption would improve clarity for the reader.

I also think the discussion should mention that the individuals in the cohort are also hiv seronegative and discuss the bias that may arise from this restriction.

Reviewer #3: Review for "Prevalence and incidence of sexually transmitted infections among South African women initiating injectable and long-acting contraceptives"

PONE-D-23-13490

Reviewer: Tim Lucas, University of Leicester, UK

Date: 2023-07-03

Overview

-----------

This paper presents data and analyses of incidence and prevalence of a number of STIs (not including HIV) in South Africa.

I have no expertise in STIs, but have been asked to review this paper as a statistician, so I will keep my comments within that area.

Overall the analyses seem sensible and the paper is well written.

I have a number of fairly minor points that I think could be clarified in the paper.

Minor Comments

---------------

My most pressing concern is the way in which the multiple-testing adjusted p values have been presented (though I would like to thank the authors for taking this issue seriously throughout the analysis).

Given that the procedure asks for all p values to be sorted, and then adjusted according to their rank, can I confirm how this has been conducted.

Has every single p value in the entire paper been put into one long list and ranked?

Or is each table of p values ranked and adjusted separately?

I think either is fine, I would just like it to be clear in the paper.

Furthermore, I think it might help interpretation if both the adjusted and unadjusted p values were presented (though I realise this might not always be easy without making the tables quite complex).

Given that the smallest raw p values are adjusted more than the bigger p values, it could well be case that the variables with the smallest adjusted p values are not the variables with the smallest unadjusted p values.

I think triangulating these two pieces of information would be useful.

The reference given for this procedure (Columb and Sagadai 2006) has a typo in the FDR step-down procedure. In step 2 they say P' = P(m / (m - (m - 1))). However, m - (m - 1) in the denominator is just equal to 1 so there's clearly a mistake. I'm not sure what the correct expression is though. But if you have directly coded up this procedure based on the equations in this paper, you will need to find the right equation and recalculate.

I would appreciate some more information in the paper about what effect the treatment/acquired immunity for these STIs has on subsequent infection and the exact ordering of treatments. Do these diseases confer immunity? You state that the baseline time is immediately before contraceptive method initiation. Were participants treated for existing STIs, and then contraception was initiated afterwards? Or was treatment initiated at the same time as contraception? If treatment was initiated at the same time as contraception this would cause some issues as presumably a course of antibiotics would be protective against reinfection.

Is there any expectation that the implementation of contraception would directly prevent sexual activity due to discomfort or due to directions from the health workers? And importantly, is this effect different across the contraception methods?

A note clarifying the role of herpes in the study would be useful. It is measured at baseline, and then not treated (as there is no treatment) and not measured in follow up. This all makes sense given the very different epidemiology of herpes compared to some of the other diseases, but I found it a bit confusing while reading the paper working out exactly what was happening.

You have tested whether the different methods of contraception yield different incidences. I might have missed it, but some statements about why you are testing this would be useful. As above, I have no expertise in this area, but I wouldn't expect these methods of contraception to alter STI infection rates. So is this just to confirm this fact? Or is there a hypothesis that the contraception methods might alter behaviour in different ways? Or something else.

Figure 2 is quite low resolution and hard to read.

6. PLOS authors have the option to publish the peer review history of their article (what does this mean?). If published, this will include your full peer review and any attached files.

Reviewer #1: No

Reviewer #2: No

Reviewer #3: **Yes: **Tim Lucas

---

## [Author Response · Author response to Decision Letter 0]

26 Sep 2023

Harryparsad et al. Manuscript Response to Editor Comments

PONE-D-23-13490R1 

Response: The research reported in this publication was supported by the Eunice Kennedy Shriver National Institute of Child Health & Human Development of the National Institute of Health under Award Number R01HD096937-01 (PIs: Morrison, Deese and Masson). The content is solely the responsibility of the authors and does not necessarily represent the official views of the National Institutes of Health. LM was supported by the Carnegie Corporation of New York, South African National Research Foundation and Australian National Health and Medical Research Council. The Evidence for Contraceptive Options and HIV Outcomes (ECHO) Trial was supported by the combined generous support of the Bill & Melinda Gates Foundation [grant OPP1032115]; the American people through the United States Agency for International Development [grant AID-OAA-A-15-00045] the Swedish International Development Cooperation Agency [grant 2017/762965-0]; the South Africa Medical Research Council; and the United Nations Population Fund. Contraceptive supplies were donated by the Government of South Africa and US Agency for International Development.

Response: The funders had no role in study design, data collection and analysis, decision to publish, or preparation of the manuscript

Response: LM, MS, PLC, XG, CML, FLDH, JS, JS, KN, KH, MB, GB, CM and JD received salary from the National Institute of Health.

RH, BM, OT received student stipends from the National Institute of Health. 

LM also received salary from the Carnegie Corporation of New York, South African National Research Foundation and Australian National Health and Medical Research Council.

Response: N/A

Response: We apologise for this and have now provided the data as Figures S3, S4 and S5.

3. Please include a complete copy of PLOS’ questionnaire on inclusivity in global research in your revised manuscript. Our policy for research in this area aims to improve transparency in the reporting of research performed outside of researchers’ own country or community. The policy applies to researchers who have travelled to a different country to conduct research, research with Indigenous populations or their lands, and research on cultural artefacts. The questionnaire can also be requested at the journal’s discretion for any other submissions, even if these conditions are not met. Please find more information on the policy and a link to download a blank copy of the questionnaire here: https://aus01.safelinks.protection.outlook.com/?url=https%3A%2F%2Fjournals.plos.org%2Fplosone%2Fs%2Fbest-practices-in-research-reporting&data=05%7C01%7CLindi.Masson%40burnet.edu.au%7C44baf172456a4dc60d1108dbb0d70be1%7Cc471a057a4e746c596342e95c39663fa%7C0%7C0%7C638298209897733676%7CUnknown%7CTWFpbGZsb3d8eyJWIjoiMC4wLjAwMDAiLCJQIjoiV2luMzIiLCJBTiI6Ik1haWwiLCJXVCI6Mn0%3D%7C3000%7C%7C%7C&sdata=eAoTSX40H8mf6ERJt3Uyp11YS6ZXEpY7Mt8dfIhi%2FbM%3D&reserved=0. Please upload a completed version of your questionnaire as Supporting Information when you resubmit your manuscript.

Response: We have uploaded this questionnaire as requested.

4. I note that in your Data Availability statement you state: “The data that support the findings of this study are available from the corresponding author upon reasonable request."

Please note that in the interest of long-term data availability, PLOS data policy states that it is not acceptable for an author to be the sole named individual responsible for ensuring data access (https://aus01.safelinks.protection.outlook.com/?url=https%3A%2F%2Fjournals.plos.org%2Fplosone%2Fs%2Fdata-availability%23loc-acceptable-data-access-restrictions&data=05%7C01%7CLindi.Masson%40burnet.edu.au%7C44baf172456a4dc60d1108dbb0d70be1%7Cc471a057a4e746c596342e95c39663fa%7C0%7C0%7C638298209897733676%7CUnknown%7CTWFpbGZsb3d8eyJWIjoiMC4wLjAwMDAiLCJQIjoiV2luMzIiLCJBTiI6Ik1haWwiLCJXVCI6Mn0%3D%7C3000%7C%7C%7C&sdata=FMXOvN2Mannty31us3pGQc7SwKPnBQjzI9h9%2B7e2iK0%3D&reserved=0).

You can, however, still have the data available upon request if you provide a non-author contact (e.g., a Research Ethics Committee or Institutional Review Board contact). You can update your Data Availability statement in your cover letter when you resubmit your revised manuscript, as we will not be able to do it whilst the manuscript is still with you or in your account.

Response: We have now included all data as Table S5.

---

## [Editor Report · Decision Letter 1]

19 Oct 2023

PONE-D-23-13490R1Prevalence and incidence of sexually transmitted infections among South African women initiating injectable and long-acting contraceptivesPLOS ONE

Dear Dr. Masson,

Thank you for submitting your manuscript to PLOS ONE. After careful consideration, we feel that it has merit but does not fully meet PLOS ONE’s publication criteria as it currently stands. Therefore, we invite you to submit a revised version of the manuscript that addresses the points raised during the review process.

You have submitted a copy of the manuscript with tracked changes, but most of the changes you state in the response sheet that you have been made, are not highlighted in the tracked changes file. This makes it difficult to evaluate e.g., final response to Reviewer #2, you state that you have added a paragraph to the Discussion, but it is not pasted into the Response Sheet or highlighted in the manuscript. Please upload a copy of the manuscript with all changes appropriately highlighted.

Reviewer #3 recommends that you present unadjusted and adjusted p-values, but you make no response to this.

PSA testing as a test for semen detection should be made clear in the Methods section - currently it's only made clear in the Discussion.

We look forward to receiving your revised manuscript.

Kind regards,

Rebecca F. Baggaley

Academic Editor

PLOS ONE

---

## [Author Response · Author response to Decision Letter 1]

26 Oct 2023

Harryparsad et al. Manuscript Response to Reviewer Comments

PONE-D-23-13490 

We sincerely appreciate the valuable comments from the Reviewers, which have helped us to improve the quality of this manuscript. The detailed responses to the Reviewers' comments are provided below.

Reviewer reports: 

Reviewer #1: This is an interesting study finding an alarmingly high prevalence and incidence of STIs among a population highly vulnerable to HIV. I think changes need to be made to the structure and the methodology before it is published. I will gladly review it again more thoroughly once my main concerns have been addressed. 

Response: We thank the Reviewer for their helpful comments. We have made the suggested changes below.

These are: 

1. I find it unclear what the aim of the analysis is. Is it to estimate STI prevalence and incidence in this population, or is it to explore the possible relationship between type of contraceptive and STI incidence? Please make the aim clear, explicitly with an aim statement at the end of the Introduction and implicitly throughout the narrative of the paper.

Response: We have added in an aim statement sentence to clarify the aim of the study (page 4, lines 17-8). 

2. I would request that a statistician also review the paper. I have a number of concerns:

a. Incidence should be defined 

b. The reason for using keplan meier and the benefit of that approach is unclear to me

c. The reason for plotting the age groups separately is also unclear. Is it to test a hypothesis that incidence would differ by age? The age categories seem somewhat arbitrary. How were they decided upon?

Response: 

a) A sentence defining incidence and prevalence was added (Page 8, lines 11-13).

b) The Kaplan-Meier method is frequently used to measure the incidence of STIs and other infections and to evaluate differences between groups. This method was chosen as follow-up time varied between participants - the Kaplan-Meier method makes appropriate allowances for censoring of observations and also makes use of the information from these subjects up to the time when they are censored. 

c) We have described the findings of previous studies in the Introduction, including the results of a meta-analysis showing that STI prevalence was higher amongst younger (18-24 years) compared to older women (25-49 years) in South Africa (9). Furthermore, differences within the younger age group have been noted, including the findings that women aged 20-24 years were twice as likely to be infected with a curable STI compared to women aged 15-19 years old (4). Therefore, we aimed to investigate these findings in our cohort and as such defined younger, middle and older age groups in a way that would approximately balance the number of participants in each group. We have clarified the grouping in the Methods section as follows: “Women were grouped by age to include approximately equal numbers within each group (18-21; 22-25; 26-33 years).”

The authors appreciate the recommendation to have our manuscript evaluated by a statistician. Reviewer 3 of this manuscript is a statistician and the critique submitted by this reviewer focused mainly on statistical issues, which we have thoroughly addressed in our revisions (see below). 

General language suggestions: 

There are too many accronyms. As a minimum, I suggest referring to the MRU site as the eThekwini site instead, and the SRC as the Tschwane site. 

Define wy when first used. 

Response: We have removed unnecessary acronyms, including MRU, SRC, TP, SA, BARC and BV as suggested and defined wy (page 2, line 17 and page 10, line 11). 

I’m not sure this makes sense: ‘seeking effective contraception with no medical contraindications to trial contraceptive methods’.

Response: We have modified this statement as follows: 

“Women enrolled were nonpregnant, HIV-seronegative, aged 16-35 years, sexually active, seeking effective contraception, had no medical contraindications to the contraceptive methods included in the trial, and with no reported use of injectable, intrauterine or implantable contraception in the past 6 months.”

Reviewer #2: Very good piece of research. I have a few minor comments. 

In the introduction I would reccomend changing the statements to "can cause shedding" and "can affect fertility" as it is not true in 100 percent of cases. I also reccomend splitting up the long list of reasons why women may be more susceptible to acquiring an sti. The list is long and the references need to match the risk factors presented. Also categories such as "gender inequality" is too broad. It is important to be more specific as to how these risks arise and why they are specific to young women.

Response: We have made the changes as suggested (pages 3-4, lines 20-24 and 1-3). We have split up the long sentences, adding in the references to the risks provided and changed “gender inequality” to “relationship power inequity” (page 3, line 23). 

There are some formatting issues in the table where the rows are not vertically aligned. And I think each table needs a description of the p-value definition. Particularly table 3 which has a "P-value" column alongside three columns under the heading "p-value". Descriptions in the text or table caption would improve clarity for the reader.

Response: We have formatted the tables (Tables 1, 2 and 3) and added the p-value definition below each table (pages 12, 13 and 14). 

I also think the discussion should mention that the individuals in the cohort are also hiv seronegative and discuss the bias that may arise from this restriction.

Response: We have added in a paragraph to discuss the above (page 17, lines 13-17).

Reviewer #3: This paper presents data and analyses of incidence and prevalence of a number of STIs (not including HIV) in South Africa. 

I have no expertise in STIs, but have been asked to review this paper as a statistician, so I will keep my comments within that area. 

Overall the analyses seem sensible and the paper is well written. 

I have a number of fairly minor points that I think could be clarified in the paper. 

Minor Comments 

---------------

My most pressing concern is the way in which the multiple-testing adjusted p values have been presented (though I would like to thank the authors for taking this issue seriously throughout the analysis). 

Given that the procedure asks for all p values to be sorted, and then adjusted according to their rank, can I confirm how this has been conducted. 

Has every single p value in the entire paper been put into one long list and ranked? 

Or is each table of p values ranked and adjusted separately? 

I think either is fine, I would just like it to be clear in the paper. 

Furthermore, I think it might help interpretation if both the adjusted and unadjusted p values were presented (though I realise this might not always be easy without making the tables quite complex). 

Given that the smallest raw p values are adjusted more than the bigger p values, it could well be case that the variables with the smallest adjusted p values are not the variables with the smallest unadjusted p values. 

I think triangulating these two pieces of information would be useful. 

The reference given for this procedure (Columb and Sagadai 2006) has a typo in the FDR step-down procedure. In step 2 they say P' = P(m / (m - (m - 1))). However, m - (m - 1) in the denominator is just equal to 1 so there's clearly a mistake. I'm not sure what the correct expression is though. But if you have directly coded up this procedure based on the equations in this paper, you will need to find the right equation and recalculate.

Response: We followed reviewer three’s advice and used Bonferroni correction instead of FDR step-down procedure for our analysis. This has been detailed in the Methods section and an appropriate reference provided. The p-values presented in the Tables are unadjusted p-values and, due to space limitations and the complexity of the tables, we have indicated the p-values that remained significant after adjusting for multiple comparisons with asterisks and have defined this in the subscripts. We have done the same for the Figures.

I would appreciate some more information in the paper about what effect the treatment/acquired immunity for these STIs has on subsequent infection and the exact ordering of treatments. Do these diseases confer immunity? You state that the baseline time is immediately before contraceptive method initiation. Were participants treated for existing STIs, and then contraception was initiated afterwards? Or was treatment initiated at the same time as contraception? If treatment was initiated at the same time as contraception this would cause some issues as presumably a course of antibiotics would be protective against reinfection. 

Response: Neisseria gonorrhoeae, Chlamydia trachomatis, Trichomonas vaginalis and Mycoplasma genitalium are all treatable with antibiotics. N. gonorrhoeae has developed resistance to antibiotics and is able to evade the immune response thereby resulting in repeated infections. C. trachomatis treatment is effective in reducing persistent and recurrent infections and is more effective if sexual partners are treated as well. Chlamydia has also been spontaneously cleared in some cases, believed to be immune induced, but re-infection is still possible. M. genitalium has also developed resistance to certain antibiotics, but re-infection is still possible. The Herpes simplex viruses 1 & 2 establish latent infection and reactivate frequently, it is therefore considered a lifelong infection. Current treatment regimens are highly effective but are unable to eliminate latent infection. T. vaginalis treatment has been effective, but previous exposure has not proven to be 100% effective and re-infection may occur (Waugh M, Gross G, Tyring SK. Sexually Transmitted Infections and Sexually Transmitted Diseases. 2011). 

We have included the following in the Discussion section (page 15, lines 20-24):

“Previous studies have hypothesized that long-lasting partial immunity to CT may develop (27) and this may explain the difference in prevalence and incidence observed between age groups. On the other hand, more frequent sexual activity reported by older women could account for the greater incidence of TV.”

Participants were screened and treated for existing STIs before contraceptive initiation as detailed in the Methods as follows: 

“Syndromic management was provided at screening; participants with positive CT/NG results not treated at screening were recalled for treatment when results became available. 

Is there any expectation that the implementation of contraception would directly prevent sexual activity due to discomfort or due to directions from the health workers? And importantly, is this effect different across the contraception methods? 

Response: We have included the following in the Discussion: 

“We did not find any association between PSA positivity across all visits and STI incidence, and we did not observe significant differences in PSA detection between age groups, study sites or contraceptive arms. However, a study by Deese et al. (28), investigating PSA detection frequency in another sub-study of the ECHO trial showed that PSA was detected less frequently in women randomised to DMPA-IM compared to copper-IUD and LNG implant. The data suggested that women using copper-IUD and LNG implant may have had condomless vaginal sex more frequently than women using DMPA-IM.”

A note clarifying the role of herpes in the study would be useful. It is measured at baseline, and then not treated (as there is no treatment) and not measured in follow up. This all makes sense given the very different epidemiology of herpes compared to some of the other diseases, but I found it a bit confusing while reading the paper working out exactly what was happening.

Response: HSV-2 serologic testing was conducted at screening and final visits as part of the parent trial, while our sub-study only evaluated active STIs, including HSV-1 and HSV-2 shedding. 

We have clarified this in the Methods section as follows: 

“Additionally, HSV-2 serologic testing was conducted at screening and final visits as part of the parent trial at Bio Analytical Research Corporation South Africa (22).” “Additional testing for active STIs was conducted as part of the present sub-study as follows.”

You have tested whether the different methods of contraception yield different incidences. I might have missed it, but some statements about why you are testing this would be useful. As above, I have no expertise in this area, but I wouldn't expect these methods of contraception to alter STI infection rates. So is this just to confirm this fact? Or is there a hypothesis that the contraception methods might alter behaviour in different ways? Or something else. 

Response: We have included the following in the Introduction: “However, contraceptive use alters the vaginal microbiome (18-20) and a non-optimal vaginal microbiome has been associated with increased risk of STIs (21).”

Figure 2 is quite low resolution and hard to read.

Response: We have made edits to ensure Figure 2 is high res and legible.

---

## [Editor Report · Decision Letter 2]

30 Oct 2023

Prevalence and incidence of sexually transmitted infections among South African women initiating injectable and long-acting contraceptives

PONE-D-23-13490R2

Dear Dr. Masson,

We’re pleased to inform you that your manuscript has been judged scientifically suitable for publication and will be formally accepted for publication once it meets all outstanding technical requirements.

Kind regards,

Rebecca F. Baggaley

Academic Editor

PLOS ONE
---

## [Editor Report · Acceptance letter]

3 Nov 2023

PONE-D-23-13490R2 

Prevalence and incidence of sexually transmitted infections among South African women initiating injectable and long-acting contraceptives 

Dear Dr. Masson:

I'm pleased to inform you that your manuscript has been deemed suitable for publication in PLOS ONE. Congratulations! Your manuscript is now with our production department. 

Kind regards, 

on behalf of

Dr. Rebecca F. Baggaley 

Academic Editor

PLOS ONE